# InGaN/GaN Distributed Feedback Laser Diodes with Surface Gratings and Sidewall Gratings

**DOI:** 10.3390/mi10100699

**Published:** 2019-10-14

**Authors:** Zejia Deng, Junze Li, Mingle Liao, Wuze Xie, Siyuan Luo

**Affiliations:** 1Microsystems and Terahertz Research Center, China Academy of Engineering Physics, Chengdu 610200, China; dengzejia@mtrc.ac.cn (Z.D.); liaomingle@mtrc.ac.cn (M.L.); xiewuze@mtrc.ac.cn (W.X.); luosiyuan@mtrc.ac.cn (S.L.); 2Institute of Electronic Engineering, China Academy of Engineering Physics, Mianyang 621999, China

**Keywords:** GaN laser diode, distributed feedback (DFB), surface gratings, sidewall gratings

## Abstract

A variety of potential applications such as visible light communications require laser sources with a narrow linewidth and a single wavelength emission in the blue light region. The gallium nitride (GaN)-based distributed feedback laser diode (DFB-LD) is a promising light source that meets these requirements. Here, we present GaN DFB-LDs that share growth and fabrication processes and have surface gratings and sidewall gratings on the same epitaxial substrate, which makes LDs with different structures comparable. By electrical pulse pumping, single-peak emissions at 398.5 and 399.95 nm with a full width at half maximum (FWHM) of 0.32 and 0.23 nm were achieved, respectively. The surface and sidewall gratings were fabricated alongside the p-contact metal stripe by electrical beam lithography and inductively coupled plasma etching. DFB LDs with 2.5 μm ridge width exhibit a smaller FWHM than those with 5 and 10 μm ridge widths, indicating that the narrow ridge width is favorable for the narrowing of the line width of the DFB LD. The slope efficiency of the DFB LD with sidewall gratings is higher than that of surface grating DFB LDs with the same ridge width and period of gratings. Our experiment may provide a reliable and simple approach for optimizing gratings and GaN DFB-LDs.

## 1. Introduction

Gallium nitride (GaN) laser diodes (LDs) are potentially used in displays, medical application, visible light communications (VLC), etc [1,2,3,4]. Some applications require a single peak and narrow-linewidth laser source. Distributed feedback lasers diodes (DFB-LDs) with these advantages have attracted extensive concern in both academia and the industry. In optical atomic clocks, an extremely narrow-linewidth blue laser is required to aim at atomic cooling transition [5]. In medical diagnostics, the DFB LD is a promising light source for fluorescence spectroscopy, where the emission wavelength can be precisely targeted [6].

GaN-based DFB LDs have been achieved by buried gratings [7,8,9], sidewall gratings, and surface gratings [10,11,12,13]. In the beginning, the first order DFB LD was achieved by establishing the buried gratings [14]. Subsequently, the third and 39th DFB LDs were fabricated by etching sidewall gratings [15,16,17]. Additionally, the single longitudinal mode emissions of the 10th DFB LDs with surface gratings were realized under optical pumping and electrical impulse driving [18,19]. In contrast to burying gratings, DFB LDs with sidewall and surface gratings do not require high-cost and hard crystal regrowth processing, preventing the device from being damaged in the regrowth process [20,21]. Surface gratings and sidewall gratings are preferred for DFB LDs due to simpler fabrication processes. Though there have been varieties of DFB LDs fabricated using surface gratings or sidewall gratings, to the best of our knowledge, all of them, regardless of grating type, are achieved on different epitaxial wafers; there are no reports on GaN DFB-LDs that share growth and fabrication processes and have surface gratings and sidewall gratings on the same epitaxial substrate. Therefore, it is inevitable that there are distinct differences in the optical and electrical properties of DFB LDs with different kinds of grating structures. It is difficult to systematically compare and analyze the performance of DFB LDs with surface and sidewall gratings, because they are fabricated on diverse epitaxial wafers. Hence, in order to further understand the performance differences of DFB LDs with different structures, it is necessary to fabricate DFB LDs with different types of gratings on the same epitaxial substrate, which can provide some experimental evidences for the optimal design of DFB LDs.

In this work, the finite-difference-time-domain (FDTD) tool was used to analyze the influence of the width of the ridge and the types of gratings on the properties of DFB LDs. Based on the designed structure of LDs, DFB-LDs with surface and sidewall gratings of different periods and ridge widths were fabricated on the same epitaxial wafer. Fabry–Pérot (F–P) LDs with the same ridge widths were also fabricated on the same epi-wafer for comparison. The well-proportioned gratings were defined alongside the p-contact metal stripe by electrical beam lithography (EBL) and inductively coupled plasma (ICP) etching. The morphology characteristics of all the LDs were observed by a scanning electron microscope (SEM) with an FEI Nova NanoSEM 450. The electrical and optical characteristics of the DFB LDs, including the linewidth, slope efficiency, and threshold current, were measured and analyzed.

## 2. Simulation and Fabrication 

According to the coupled mode theory, the coupling coefficient can be achieved approximately by the following equation [22]:(1)$$\kappa =\frac{2\left(n_{2}-n_{1}\right)}{\lambda_{1}}\frac{\sin\left(\pi \;m\;\gamma \right)}{m}$$
where *m* is the grating order, *λ*_1_ is the wavelength, *n*_1_ is the effective modal index in the broad area including the sections of the ridge and grating, *n*_2_ is the effective modal index in the section of the ridge, and *γ* is the duty-cycle of gratings.

The effective modal index (*n*_eff_) of the waveguide in a GaN LD can be calculated using the finite-difference-time-domain (FDTD), where the epitaxial structure and device structure of the simulation can be found in the parameters displayed in the following sections. Generally, there is a decreasing trend in modal index with ridge width at a certain etch depth. Typical optical field distributions are given in Figure 1, with ridge widths of 2.5 and 5 μm. It is obvious that a reduction of the waveguide width restriction the optical field more, resulting in a decreasing in mode number. 

The refractive modal index increases with ridge width and gradually approaches the maximum value when the ridge width is relatively large, which is shown in Figure 2a. It is worthy pointing out that coupling length *ĸL* has a great influence on the characteristics of the DFB LDs. Figure 2b shows the coupling length *ĸL* as a function of the ridge width when the etch depth was 500 nm and the cavity length was 600 μm. For the sidewall grating, the broad section including the ridge and grating had a width twice the size of the section of the ridge, whereas for the surface grating, the broad section possessed a fixed width of 80 μm. Obviously, *ĸL* showed a decreasing trend as the ridge width increased. It must be pointed out that this calculation was based on the fundamental mode; in the high order, mode *ĸL* had a relatively large value, leading to a moderate reflectivity. Since the surface gratings had a greater width (and thus refractive index) in the broad area for the same ridge width (compared with the sidewall gratings), they had a bigger refractive index difference Δ*n* = (*n*_2_ − *n*_1_). As demonstrated in Equation (1), the coupling length *ĸL* was proportional to refractive index difference Δn when the order and duty ratio of the gratings were defined. As such, the *ĸL* of the DFB LD with surface gratings was slightly higher than that of the one with sidewall gratings when the ridge widths of the two kinds of structures were identical.

According to the Bragg condition, *Λ = λm/2n*_eff_ and the characterization of the emission at around 403 nm of the F–P LDs fabricated on the epitaxial wafer, the periods of the 10th-order and 20th-order Bragg grating were calculated and found to be 824 and 1648 nm, respectively. Another parameter for the gratings is the duty cycle *γ = (Λ − o)/Λ*, where *o* denotes the width of the etched trench. A duty cycle of around 0.8 was adjusted for the gratings.

The designed complete structures of the DFB LDs with surface and sidewall gratings are depicted in Figure 3. It is worth noting that the surface gratings had a larger grating width than the sidewall grating and were much wider than the width of the ridge waveguide, while the width of the sidewall grating was closer to that of the ridge waveguide. The LDs were fabricated from the AlInGaN epi-structure composed of n-type and p-type epitaxial layers and InGaN/GaN multiple quantum wells (MQWs) designed for the emission of 403 nm, including a lower cladding layer of 850 nm Al_0.075_Ga_0.925_N:Si, a 60-nm-thick In_0.03_Ga_0.97_N:Si lower waveguide layer, a multiple quantum well with three 6.5- or 2-nm-thick undoped GaN quantum barriers and two 2.7 nm In_0.1_Ga_0.9_N wells, an upper waveguide layer of doped 60 nm InGaN:Mg, a 20 nm Al_0.13_G_0.87_N:Mg electron blocking layer, a 450 nm Al_0.05_Ga_0.95_N:Mg upper cladding layer, and a 10 nm GaN:Mg subcontact layer. The F–P LDs and DFB LDs were fabricated on the same epi-wafer by the same manufacturing process except for the building of the surface and sidewall gratings of DFB LDs. The 2.5-, 5- and 10-μm-wide Ni/Au p-contact stripes with a thickness of 10/30 nm were formed on the wafer by electron beam evaporation and a rapid thermal process. The 150-nm-thick SiO_2_ layer was deposited on the whole wafer by plasma-enhanced chemical vapor deposition (PECVD) as the hard mask of etching of gratings and ridge waveguides for DFB LDs. The 250-nm-thick positive polymethyl methacrylate (PMMA) resist was used to define the gratings with periods of 824 and 1648 nm, the patterns of the gratings were constructed on both sides of the p-contact stripes and perpendicular to them with this method of electron beam lithography (EBL), the model of the EBL was Raith i-line plus with a write resolution of 7 nm. The patterns of gratings and ridge waveguides were transferred to the epi-structure of the wafer through reactive ion etching based on CHF_3_/SF_6_ gas and the dry etching of inductively coupled plasma (ICP) using Cl_2_/BCl_3_ gas in succession. The 300 nm-thick SiO_2_ insulating layer was formed on the wafer by PECVD, and the 2-, 4.5- and 8.5-μm-wide openings on the ridge were given shape by dry etching the SiO_2_ layer using CHF_3_/SF_6_ mixed gas before the 50/250 nm Ti/Au metal pad was deposited. The back side of the wafer was thinned and polished to a thickness of 120 μm, and then an n-contact Ti/Pt/Au metal layer with a thickness of 50/50/250 nm was deposited on it. The processed sample was cut into bars with a cavity length of 600 μm, and both front and back facets were uncoated.

## 3. Results and Discussions

### 3.1. The DFB LDs with the Surface Gratings 

Table 1 compares the structure parameters of different manufactured LDs. Devices 1 and 4 are F–P LDs with ridge widths of 2.5 and 10 μm, respectively, that are compared with the fabricated DFB-LDs with surface gratings. Devices 2 and 3 have the same ridge width and width and duty ratio of surface gratings, but they have different grating periods of 824 and 1648 nm. The same parameters also apply to Devices 5 and 6 except for the ridge width. All of them were fabricated from the same epitaxial wafer and shared the same processing.

The structures of the ridge and gratings of the DFB LDs with surface gratings were well formed, as shown in Figure 4. The high resolution SEM graph in Figure 4a shows a lateral view of the DFB LD with surface gratings, thus displaying the general structural features of the device. The actual parameters of the ridge and the grating of the processed DFB LD are shown in Figure 4b, which were close to the designed values shown in Table 1. Figure 4c presents the sectional feature of the surface gratings with a period of 1648 nm. It can be concluded that the gratings have the desired quality and a depth of 481 nm close to the head of the upper waveguide layer, which means that they basically meet our design requirements.

The spectral measurements were conducted by a fiber spectrum analyzer (BIM6002, Brolight, Hangzhou, China) with a resolution of 0.16 nm under a pulsed operation of a 500 ns pulse length and 1kHz repetition frequency. Figure 5 shows the emission spectrum chart of the DFB LDs (Devices 1–6) with surface gratings and F–P LDs. All of them were operated at the impulsive condition of 1.2 times the threshold current by the fiber spectrometer. Since the emission peaks of those LDs had different intensities under certain test currents, the differences in the position of the bottom of the spectrums of LDs were caused by the normalization process of dividing by the different highest intensities. The emission peaks of Devices 1–6 were located at 402.13, 398.86, 398.50, 402.86, 400.50 and 400.86 nm, respectively. Obviously, the DFB LDs had a shorter lasing wavelength than the F–P LDs with the same ridge width because of the interaction between gratings and multiple quantum wells. The offset of the 2~3 nm emission wavelength between the DFB and the F–P laser also confirms the modulation effect of the grating. The full width at half maxima (FWHM) corresponding to DFB LDs for Devices 2, 3, 5, and 6 were 0.37, 0.32, 0.52, and 0.50 nm, respectively. We can conclude that the DFB LDs exhibited a narrower emission width and a single peak emission because of the modulation of the gratings, while the F–P LDs for Devices 1 and 4 had distinct multimode characteristics. Additionally, the FWHM of the emission of the DFB LD with the same ridge seems to have rarely been related to the period of the gratings, while the DFB LD with the 2.5 μm ridge width had a lower FWHM than that with the 10 μm ridge width. This can be partly explained by the fact that, as depicted in Figure 1, the simultaneous oscillation of lateral modes sharing the same longitudinal mode number eventually results in a wider spectral width. In addition, according to the equation *L = λm*/2*n*_eff_—where *n*_eff_ ≈ 2.5 and *L* = 600 μm—the expected free spectral range (FSR) of the longitudinal modes of the F–P laser could be approximately 0.05 nm at around 400 nm. Therefore, the DFB LDs possibly showed multi-mode operation, but this phenomenon was not been confirmed because of the lack of the measurement condition of the high resolution spectrum and the lateral far field. 

### 3.2. The DFB LDs with the Sidewall Gratings

In addition, the DFB LDs with sidewall gratings were fabricated on the same epitaxial wafer, and they shared the same processing steps with the surface grating DFB LDs. The structure parameters of the fabricated LDs are shown in Table 2. Devices 7 and 8 had the same ridge width of 2.5 μm, gratings width of 1.25 μm, and 80% duty ratio of gratings, but they had different grating periods of 824 and 1648 nm, respectively. The same parameters for DFB LDs with sidewall gratings except for the widths of the ridge and gratings were applied to Devices 10 and 11. Device 9 was the F–P LDs with a ridge width of 5 μm, which was in contrast with the DFB LDs for Devices 10 and 11.

Figure 6 shows that the DFB LD with 20th order sidewall gratings was well fabricated. Figure 6a is the top view of the DFB LD with sidewall gratings, and it depicts the structure of ridge, sidewall gratings, double etching grooves, and the p-contact metal on the ridge. Figure 6b,c shows the actual parameters of the ridge and gratings of the processed device, which are basically in conformity to the values in Table 2. Sidewall gratings with a period of 1648 nm, a duty ratio of approximately 80%, and a width of 1.25 μm of each side alongside the ridge waveguide were observed. Since the corresponding etchings processes were under the same experimental condition, the surface and sidewall gratings had a similar etching depth and sectional morphology, as shown in Figure 4. The notches of the gratings had a tilt angle as a result of the ICP etch process, so the average duty ratio of the fabricated gratings was greater than 80% [23,24]. Moreover, the simulation results showed that the influence of the relative large grating angle of 70° could be ignored [25], offering a high enough reflectivity for the DFB LDs. 

The emission spectrum of the DFB LDs with sidewall gratings and the F–P LDs for Devices 7–11 can be seen in Figure 7, and the LDs were operated at a current around 20% above threshold current. The emission peaks of the Devices 7–11 were 399.59, 399.95, 403.76, 401.4 and 401.77 nm, respectively. The FWHMs of the DFB LDs for Devices 7, 8, 10, and 11 were 0.25, 0.23, 0.42, and 0.48 nm, respectively, which were slightly higher than the resolution of the fiber spectrometer. In the context of comparing the multimode morphology of F–P LDs with the ridge of 5 μm during lasing, the emission spectrum of the DFB LDs showed a single peak because of the existence of the gratings. Similar to the previous results, the DFB LDs of sidewall gratings had a lower lasing wavelength than the F–P LDs with the same ridge width. In addition, the DFB LDs with a 2.5 μm ridge width had a lower FWHM than those with the 5 μm ridge width, thus indicating that a narrow ridge width is favorable for the narrowing of the linewidth of the DFB LD. According to the spectrum characteristics, we could speculate that the effect of grating periods on the line width of LDs is not very evident.

### 3.3. The Comparison of Properties of the DFB LDs with Surface and Sidewall Gratings

Meanwhile, based on the good linewidth characteristics of DFB LDs with a ridge width of 2.5 μm, the characteristics of DFB LDs with sidewall and surface gratings and the F–P LD were compared in the case that their ridge widths were all 2.5 μm. The sidewall gratings need lower costs for writing a smaller area of patterns than surface gratings which use EBL. The light output power and voltage versus current (L–I–V) characteristics of the LDs were tested under the pulsed driving condition with a pulse width of 1 μs and a pulse repetition frequency of 10 kHz at room temperature to avoid generating excess heat. Figure 8 shows L–I–V characteristics of the DFB LDs and the F–P LDs with the same ridge width of 2.5 μm. The values of the threshold currents of Devices 1, 2, 3, 7, and 8 were between 450 and 560 mA, and their slope efficiencies were approximately 0.24, 0.12, 0.17, 0.13, and 0.19 W/A, respectively. The deviations of the threshold current and slopes efficiency of the LDs with the same structures were less than 8% and 13%, respectively. However, all of the F–P and DFB LDs tested showed comparatively high threshold currents and lower slope efficiencies, which was mainly deemed to owe to the performance of the epitaxial wafer. More intuitive and detailed results are shown in Table 3. 

From the data in Figure 8 and Table 3, it can be seen that DFB LDs with the same ridge width and a lower grating period exhibited lower slope efficiency because of the higher value of *ĸL* compared to those with high order gratings [26]. In addition, the slope efficiency of DFB LDs with sidewall gratings was slightly higher than that of the DFB LDs with surface gratings but smaller than F–P LDs because of the scatter losses and diffraction losses from the gratings. Besides, the FWHM of DFB LDs with sidewall gratings was slightly lower than that of DFB LDs with surface gratings, and this may have been due to the fact that the sidewall grating DFB LD had a narrower effective width in the whole structure compared to the surface grating of the DFB LD with the same ridge width. Given its smaller grating area, the sidewall grating DFB LD had fewer transverse modes, thus resulting in a smaller FWHM compared to the surface grating DFB LD. Additionally, the linewidth of the DFB LDs with a single peak emission showed a drastic reduction compared to that of the F–P LDs with multiple lasing peaks.

## 4. Conclusions

In conclusion, the GaN-based DFB LDs that integrated with the surface and sidewall gratings were investigated by the FDTD method. The surface and sidewall gratings alongside the p-contact metal stripe on the ridge waveguide were fabricated by EBL and ICP etching on the same epitaxial wafer. The DFB LD with the 20th order, 80% duty-cycle surface gratings showed a single wavelength emission at 398.86 nm with an FWHM of 0.32 nm under the electrical pulsed driving, and the DFB LD with the 20th order, 80% duty-cycle sidewall gratings obtained a peak emission at 399.95 nm with an FWHM of 0.23 nm. Additionally, both of the DFB LDs showed a narrower linewidth compared to that of the F–P LDs. Moreover, the FWHM of the DFB LDs with the ridge width of 2.5 μm was obviously lower than that of the DFB LDs using the same type of gratings with ridge widths of 5 or 10 μm, which indicates that the narrow ridge width was favorable for the narrowing of the linewidth. Furthermore, the sidewall grating DFB LDs possessed a slightly higher slope efficiency than that of the surface grating DFB LDs with the same ridge width and period of gratings. Given that, the DFB LD with sidewall grating required a lower fabrication cost and achieved better device performance compared to the surface grating DFB LDs, which makes it a better choice for these applications. In addition, in order to further improve the performance of the DFB LDs, the optimization of the structure of the gratings and the conduction of the cavity surface coating process are required to reduce threshold current and improve slope efficiency. Additionally, the side-mode suppression ratio and the high resolution spectral measurement are also required in future work.

## Figures and Tables

**Figure 1 micromachines-10-00699-f001:**
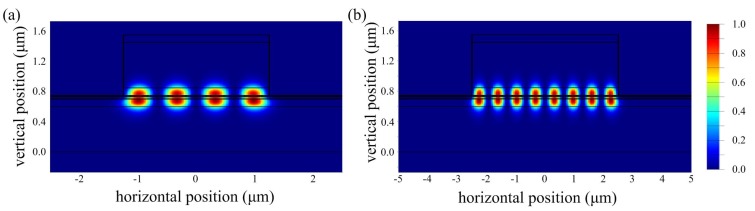
Optical field profiles for ridge widths of (**a**) 2.5 μm and (**b**) 5 μm.

**Figure 2 micromachines-10-00699-f002:**
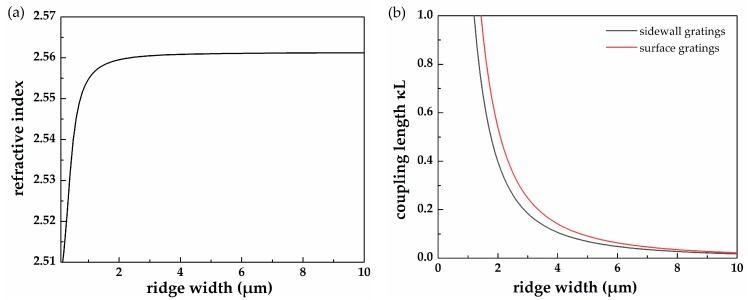
(**a**) Refractive index as a function of the ridge width. (**b**) Coupling length *ĸL* as a function of the ridge width of the sidewall and surface gratings. Etching depth: 500 nm.

**Figure 3 micromachines-10-00699-f003:**
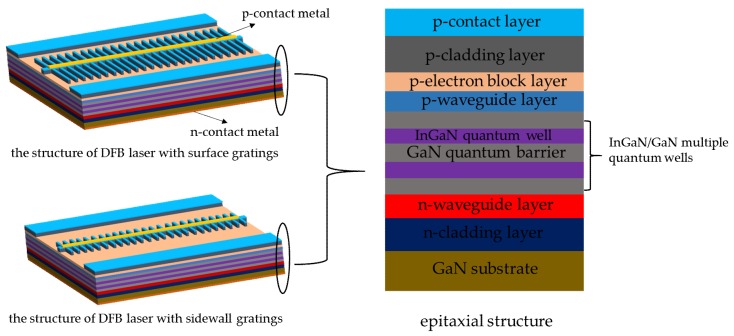
The structures of processed distributed feedback laser diodes (DFB-LDs) with surface gratings and sidewall gratings, as well as the diagrammatic drawing of the cross profile of the gallium nitride (GaN)-based epitaxial wafer.

**Figure 4 micromachines-10-00699-f004:**
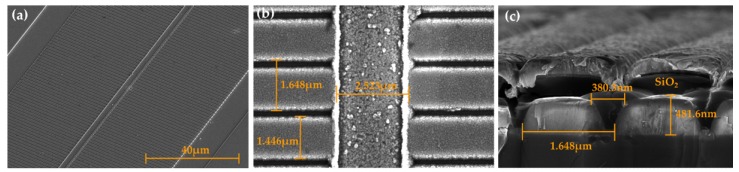
Scanning electron microscope (SEM) images of (**a**) an oblique view of the fabricated DFB laser diodes with surface gratings, (**b**) a top vision of the 20th order surface gratings alongside a metal stripe, and (**c**) a sectional structure of the surface gratings perpendicular to the facet of LDs.

**Figure 5 micromachines-10-00699-f005:**
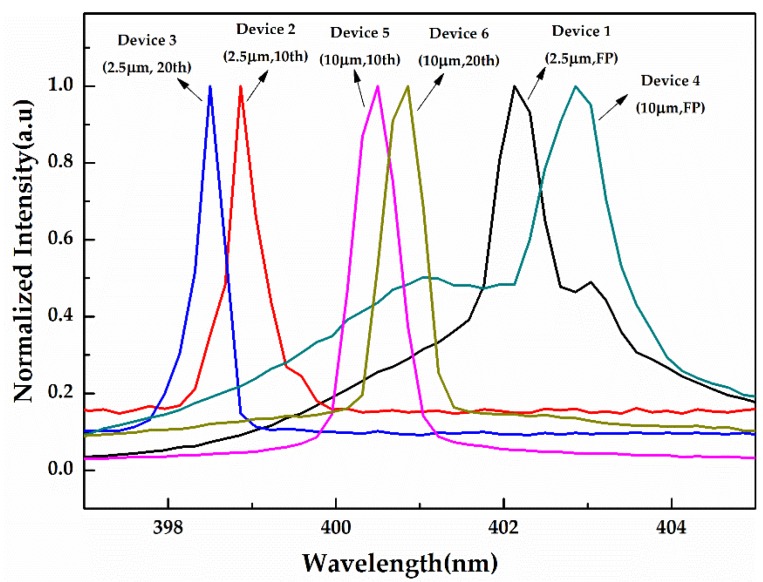
Emission spectrograms of fabricated laser diodes (Devices 1–6) driven by electrical impulses with a 500 ns pulse width and a 1 kHz repetition rate.

**Figure 6 micromachines-10-00699-f006:**
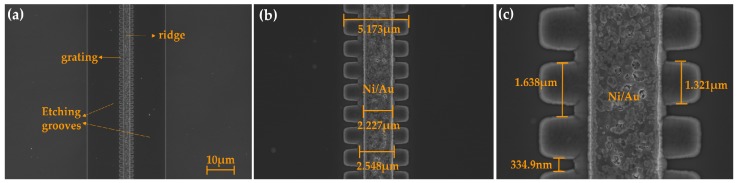
SEM images of a top view of the fabricated DFB laser diodes with 20th order sidewall gratings. (**a**) An overall view of the DFB LD with double etching grooves; (**b**) a low power and (**c**) a high power perspective of the ridge waveguide and the sidewall gratings along it.

**Figure 7 micromachines-10-00699-f007:**
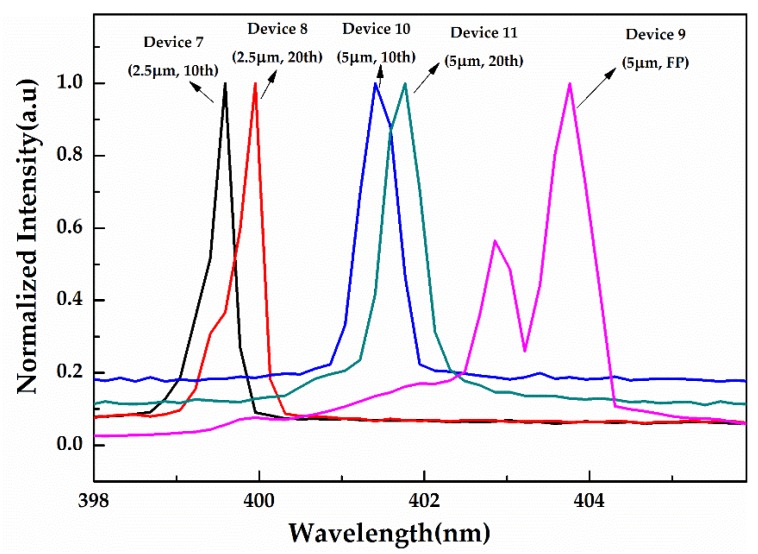
Emission spectrograms of processed laser diodes for Devices 7–11 driven by electrical impulses with a 500 ns pulse width and a 1 kHz repetition rate.

**Figure 8 micromachines-10-00699-f008:**
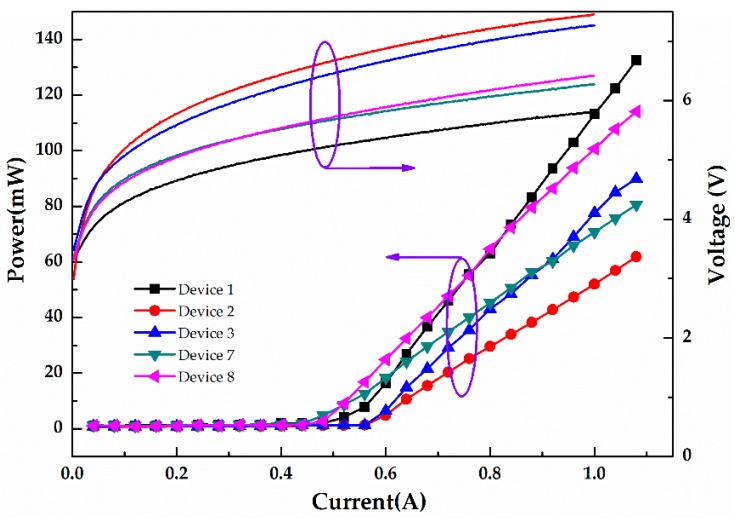
Light output power–current–voltage characteristics of Devices 1, 2, 3, 7, and 8 operated in pulsed mode with a 1 μs pulse width and a 10 kHz repetition rate.

**Table 1 micromachines-10-00699-t001:** The structure parameters of the fabricated distributed feedback laser diodes (DFB-LDs) of surface gratings and Fabry–Pérot (F–P) LDs.

Sample	Device 1	Device 2	Device 3	Device 4	Device 5	Device 6
Ridge width	2.5 μm	2.5 μm	2.5 μm	10 μm	10 μm	10 μm
Width of grating of each side	-	40 μm	40 μm	-	40 μm	40 μm
Period of gratings	z	824 nm	1648 nm	-	824 nm	1648 nm
Duty ratio of grating	-	80%	80%	-	80%	80%

**Table 2 micromachines-10-00699-t002:** Structure parameters of the fabricated DFB LDs of sidewall gratings and F–P LDs.

Sample	Device 7	Device 8	Device 9	Device 10	Device 11
Ridge width	2.5 μm	2.5 μm	5 μm	5 μm	5 μm
Width of gratings of each side	1.25 μm	1.25 μm	-	2.5 μm	2.5 μm
Period of gratings	824 nm	1648 nm	-	824 nm	1648 nm
Duty ratio of gratings	80%	80%	-	80%	80%

**Table 3 micromachines-10-00699-t003:** Electrical properties of the fabricated DFB LDs and F–P LDs.

Sample	Device 1	Device 2	Device 3	Device 7	Device 8
Type of gratings	-	Surface	Surface	Sidewall	Sidewall
Order of gratings	-	10th	20th	10th	20th
Threshold current	(518 ± 12) mA	(539 ± 4) mA	(550 ± 10) mA	(458 ± 7) mA	(485 ± 19) mA
The slopes efficiency	(235 ± 15) mW/A	(116 ± 3) mW/A	(172 ± 11) mW/A	(129 ± 7) mW/A	(189 ± 6) mW/A
FWHM	1.96 nm	0.37 nm	0.32 nm	0.25 nm	0.23 nm

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
