# Peer review of "InGaN/GaN Distributed Feedback Laser Diodes with Surface Gratings and Sidewall Gratings"

_micromachines, 2019, doi:10.3390/mi10100699_

Round 1
Reviewer 1 Report
A more descriptive discussion of the results is needed to improve the overall work:
1) More insights into the comparison between the surface gratings and sidewall gratings is required. Please elaborate on Figure 2 explaining the reason why the surface gratings kL values are higher compared to the sidewall gratings.
2) Similarly, the result comparison between the DFB and F-P structure requires some more explanation. Why does the DFB show shorter emission wavelength? The authors mention that it’s because of the interaction of gratings and multiple quantum wells. Please elaborate.
3) From Figure 8 and Table 3 the DFB (Device 2 and 3) lasers show higher threshold compared to F-P structures. Usually, due to the distributed mirrors in a DFB laser diode, the gain at the threshold is decreased so lasing should begin before in DFB mode compared to F-P mode, within the same device. Therefore the DFB laser should have an overall lower threshold.
The authors mention that the F-P laser diode shows lower threshold voltage because it does not suffer from the losses introduced by the gratings. However, Reference 15 shows that the DFB laser has lower threshold compared to the F-P due to the increased feedback from the gratings. Please comment.
4) Similarly, from Table 3: For higher coupling strength, the threshold current should decrease because of the distributed optical reflection mechanism. This is not observed in the comparison between the surface and sidewall grating devices where the surface grating device shows higher kL values at the same ridge width. Please comment.
5) A more elaborate description on the smaller FWHM of the sidewall gratings DFB compared to the surface grating design would give more insights into the result discussion.
Reviewer 2 Report
In general, the concept of comparing DFB lasers with surface and sidewall etching from the same substrate is good. However, there are a number of questions which arise.
I think a better explanation of why this work has been done needs to come across in the introduction. Is this the first time anyone has carried out a direct comparison of sidewall and surface gratings from the same substrate?
How many data points were used in Figure 2? It is my understanding that in order to get a good working DFB, kL should be ~1 so I find it hard to believe that you had a working DFB with a 10um ridge width when the coupling is almost nothing?
Given these DFBs are 10th and 20th order (even numbers), does this make them top-emission devices?
In Figure 3, it is not obvious what the difference is between the surface and the sidewall gratings images. An explanation of the different fabrication methods would help or something which differentiates the images.
Do you know the side mode suppression ratios (SMSR) for these DFBs? In Figure 5, it does not look like there are many data points and that devices 5 and 6 do not look single mode. Also, why was the spectral measurement taken under pulsed conditions and not CW? How much power was being measured? Do the DFBs stay single mode when you change the current and/or the temperature? - I have similar concerns over Figure 7.
You should include a scale bar in Figure 6(a) as the bar underneath the image is unreadable.
Overall, I'm not sure what this adds to the research community as I do not think there is much of a scientific input. Yes, the comparison of ridge widths is good, but I think in this field, most people have already ruled out the larger ridge widths. I think with a bit of work, it will be possible to have this published but I recommend major revisions.
Reviewer 3 Report
The authors present data on GaN-based DFB laser diodes with either surface gratings or sidewall gratings. The data is compared to Fabry-Pérot type ridge waveguide laser diodes. Since DFB lasers in the violet-blue-green spectral region have received some interest during the past couple of years the topic of the paper is of sufficient importance for the community.
The paper needs various major revisions for being considered for publication:
Simulation data are presented in section 2. The results can not be evaluated because basic information is missing:- the assumed heterostructure (layer compositions and thicknesses)
- the assumed geometrical structure of the grating (period, duty cycle, shape of grooves, lateral extension, material/thickness of insulator and p-metal stack, etc.) The finding of the simulations that a narrower ridge reduces the number of lateral modes is trivial and is not worth publishing. With this in mind, what is Fig. 1 good for? The narrowest structures which have been simulated and fabricated have a ridge width of 2.5 µm. Obviously the ridges are too broad to obtain lateral single mode operation. Considering the goal of a DFB laser diode to provide single longitudinal and lateral mode operation, what is the idea of studying this type of broad lasers? The facet of the devices has not been coated. This is not reasonable for DFB laser diodes where Fabry Pérot modes should be suppressed. The authors should provide either data on facet coated devices or data which show that Fabry-Pérot modes do not play a role in their DFB laser diodes. The notches etched in the semiconductor are not of rectangular shape. Still the authors simply take the surface opening of the notches to derive a duty cycle for the mode and neglect that the mode is mostly affected by the lower part of the notches. This whole issue needs to be addressed. Similar as for the simulations, the experimental heterostructure (layer compositions and thicknesses) need to be provided. The spectra provided in Fig. 5 and 6 are of very low quality considering that the authors want to use them to judge on the mode spectrum of the devices. The authors specify a resolution limit of 0.16 nm, which is unacceptably bad. Therefore, all derived FWHM numbers are of little meaning. Better resolved spectra need be provided. Also, the authors need to explain the background in many of their spectra. Fig. 8: What is the reason for the different I-V curves of the devices? Fig. 8 + Table 3: The threshold currents of all devices (including the ridge waveguide Fabry-Pérot laser) are huge. They correspond to threshold current densities of 30-37 kA/cm2. Also, the slope efficiencies are very low with 0.1-0.2 W/A. State of the art devices with emission around 400 nm should show <<3 kA/cm2 and >>1 W/A. What is the reason for this bad performance? How far data from such low-performance devices can be used to draw general conclusions on the feasibility of different grating concepts? Table 3: The authors have to provide confidence intervals for their threshold currents and slope efficiencies before comparing the numbers of devices with different designs. The english language is unsatisfactory and needs considerable polishing. The meaning of certain phrases is even unclear at all, e.g. "double etching grooves".
Round 2
Reviewer 1 Report
Please revise the english.
Author Response
Thank you for your positive evaluations and suggestion. We have made appropriate modifications to the sentence and grammar of the manuscript, we believe the modified version is just fine to readers.
Reviewer 2 Report
After the spelling mistakes are corrected and the English is improved, I believe there is enough in this paper to warrant publication.
Author Response

(The authors gave the same response as above.)

Reviewer 3 Report
The authors have commented on all my points of criticism and made corresponding corrections to the manuscript. Since not all of the corrections are satisfactory, some further revisions are required:
1) Device structures (epitaxial heterostructure, chip layout) are provided on page 4 to explain the fabricated lasers. However, simulation results are provided on page 2 which can only be rated if the device structure assumed for the simulations is known at this point. Is the simulated structure the same as the fabricated structure? In any case, the simulated structure needs to be specified at the point where the simulation results are presented.
2) The authors added a comment on the modal refractive index on page 2. Please provide the calculated numbers for the modal refractive index of the devices presented in Fig. 1.
3) The authors argue that the simulations are useful as they prove lateral multi-mode behavior which is the reason for the linewidth broadening of the DFB lasers. It's hard for me to accept this justification. The multi-mode behavior of the devices is obvious from the very beginning due to the large ridge width. The simple statement of expected multi-mode behavior does not require sophisticated simulations. Could the authors provide the theoretical spectral distribution of all stable modes in their devices and compare it to their FWHM measurements? In this case, I could see some benefit for showing simulation data in the paper.
4) The provided low resolution spectra are still unsatisfactory. However, I do understand that the authors are unable to provide better data at this point. When discussing the FWHM values on page 5, please add:
- The expected spectral distance of longitudinal modes of a Fabry Perot laser.
- The conclusion that the peak widths cover several longitudinal modes, i.e. the DFB lasers possibly show multi-mode operation not only with respect to the lateral modes but the longitudinal modes as well. High resolution spectra and measurements of the lateral far field are required for final conclusions.
5) In their reply letter the authors argue on the impact of the angle of the notches on the effective duty cycle of the grating. The authors should stress this issue not only in the reply letter but in the manuscript as well. I.e. the reader of the paper should be informed why the authors state a duty cycle of 80 % although this duty cycle is only valid for the top of the grating where the mode has its smallest intensity whereas the intensity is much larger at the bottom of the notches where the duty cycle is much larger.
6) The accuracies for the threshold current and the slope efficiency provided in Table 3 are very small, i.e. <1 %. I assume that these numbers merely reflect the accuracy of deriving the threshold current and the slope efficiency from a selected L-I curve. The numbers do not reflect the scattering of data from nominally identical devices of a wafer. The authors should concretize this issue in the manuscript.
7) The english language is still unsatisfactory at many points and needs to be improved. It has not improved with the second version of the manuscript.
